# Management of the Contralateral Neck in Unilateral Node-Positive Oral Squamous Cell Carcinoma

**DOI:** 10.3390/cancers15041088

**Published:** 2023-02-08

**Authors:** Christian Doll, Friedrich Mrosk, Lea Freund, Felix Neumann, Kilian Kreutzer, Jan Voss, Jan-Dirk Raguse, Marcus Beck, Dirk Böhmer, Kerstin Rubarth, Max Heiland, Steffen Koerdt

**Affiliations:** 1Department of Oral and Maxillofacial Surgery, Charité—Universitätsmedizin Berlin, Corporate Member of Freie Universität Berlin and Humboldt-Universität zu Berlin, Augustenburger Platz 1, 13353 Berlin, Germany; 2Berlin Institute of Health, Charité—Universitätsmedizin Berlin, Charitéplatz 1, 10117 Berlin, Germany; 3Department of Oral and Maxillofacial Surgery, Fachklinik Hornheide, Dorbaumstraße 300, 48157 Muenster, Germany; 4Department of Radiation Oncology, Charité—Universitätsmedizin Berlin, Corporate Member of Freie Universität Berlin and Humboldt-Universität zu Berlin, Augustenburger Platz 1, 13353 Berlin, Germany; 5Institute of Biometry and Clinical Epidemiology, Charité—Universitätsmedizin Berlin, Corporate Member of Freie Universität Berlin and Humboldt-Universität zu Berlin, Charitéplatz 1, 10117 Berlin, Germany; 6Institute of Medical Informatics, Charité—Universitätsmedizin Berlin, Corporate Member of Freie Universität Berlin and Humboldt-Universität zu Berlin, Charitéplatz 1, 10117 Berlin, Germany

**Keywords:** oral squamous cell carcinoma, elective neck dissection, neck management, contralateral neck, contralateral metastasis, occult metastasis, de-escalation, prognosis

## Abstract

**Simple Summary:**

Elective management of the contralateral neck in lateralized oral squamous cell carcinoma remains a matter of debate, and current guidelines provide only little guidance in this regard. Especially patients with ipsilateral cervical lymph node metastasis are of interest since several studies showed a higher risk for contralateral metastasis in these cases. The present study is a retrospective analysis of this cohort over a 10-year period at a high-volume tumor center in Germany. The results of this study show that the prevalence of occult contralateral neck metastases is low, and that contralateral elective neck dissection should not be performed routinely in this cohort.

**Abstract:**

Introduction: In lateralized oral squamous cell carcinoma (OSCC) with ipsilateral cervical lymph node metastasis (CLNM), the surgical management of the unsuspicious contralateral neck remains a matter of debate. The aim of this study was to analyze this cohort and to compare the outcomes of patients with and without contralateral elective neck dissection (END). Material and Methods: A retrospective analysis of patients with lateralized OSCC, ipsilateral CLNM (pN+) and contralateral cN0-stage was performed. Patients were divided into two groups according to the surgical management of the contralateral neck: I: END; and II: no END performed. Adjuvant radiotherapy was applied bilaterally in both groups according to individual risk. Results: A total of 65 patients (group I: 16 (24.6%); group II: 49 (75.4%)) with a median follow-up of 28 months were included. Initially, there was no case of contralateral CLNM after surgery. During follow-up, 6 (9.2%) patients presented with recurrent CLNM. In 5 of these cases (7.7%), the contralateral neck (group I: 3/16 (18.8%); group II: 2/49 (4.1%)) was affected. Increased ipsilateral lymph node ratio was associated with contralateral CLNM (*p* = 0.07). END of the contralateral side showed no significant benefit regarding OS (*p* = 0.59) and RFS (*p* = 0.19). Conclusions: Overall, the risk for occult contralateral CLNM in patients with lateralized OSCC ipsilateral CLNM is low. Our data suggest that END should not be performed routinely in this cohort. Risk-adapted radiotherapy of the contralateral neck alone seems to be sufficient from the oncological point of view.

## 1. Introduction

Oral squamous cell carcinoma (OSCC) is accompanied by cervical lymph node metastasis (CLNM), and not only in advanced stages. The presence of CLNM, however, is one of the most important prognostic factors and is associated with a poor overall prognosis [1,2,3].

Even though elective neck dissection (END) on the ipsilateral site has been established as the gold standard [4,5,6], novel approaches, such as sentinel lymph node biopsy (SLNB), are alternatives in the treatment of the clinically unsuspicious neck (cN0) [7,8]. However, with occult CLNM rates of about 30%, END is still widely recognized [9,10]. Nodal involvement of the contralateral neck in particular is associated with an extremely poor prognosis [11,12,13,14,15]. As reported by Koo et al., five-year survival rates significantly dropped from 74% in patients with no contralateral CLNM to 43% in the presence of contralateral CLNM [14]. Studies have shown that tumors involving the midline are at highest risk for contralateral CLNM [16,17]. On the other hand, lateralized OSCC presents with much lower contralateral CLNM rates, especially if the ipsilateral neck is clinically unsuspicious [17]. However, Capote-Moreno et al. identified ipsilateral CLNM as one of the most important predictors of contralateral CLNM in a retrospective study [15]. In clinical decision-making, ipsilateral CLNM may justify contralateral neck dissection. However, data on the extent of contralateral involvement are still limited, as there are several influencing factors [18]. Moreover, current guidelines on this matter are still imprecise [4,5,19]. The question is whether to address the contralateral neck surgically in patients with ipsilateral CLNM in lateralized OSCC. Therefore, the current retrospective study aimed to gain more insight into the following questions in strict unilateral OSCC: (i) in cases of ipsilateral CLNM, how likely is contralateral CLNM after initial cN0 staging of the contralateral neck side during follow-up, and (ii) is risk-adapted radiation only an adequate treatment strategy for the contralateral cN0 neck?

## 2. Material and Methods

### 2.1. Ethical Approval

The institutional review board of the Charité—Universitätsmedizin Berlin, Germany gave ethical permission for data collection and publication (EA2/077/20).

### 2.2. Study Design

In this retrospective study, we analyzed all patients with newly diagnosed OSCC who were surgically treated in a curative setting between 2010 and 2020 in the Department of Oral and Maxillofacial Surgery at the Charité—Universitätsmedizin Berlin, Germany. Clinical and radiographic follow-up data was evaluated until August 2022.

The inclusion criteria were as follows (Figure 1):
I.Histologically proven OSCC, which was unilaterally located without reaching the midline. All patients underwent ipsilateral neck dissection (ND) and presented with histologically proven ipsilateral CLNM (≥pN1).II.The contralateral neck was determined to be clinically unsuspicious (cN0).III.Patients either receiving END for the contralateral neck (group I) or no surgical treatment for the contralateral neck (group II).IV.No history of previous OSCC or radiation therapy in the head and neck area.V.All patients underwent surgery with a curative intent and were treated with bilateral adjuvant radiotherapy (RT) with or without chemotherapy according to the national guidelines [4].VI.The minimum follow-up period was 6 months.

Patient data regarding preoperative staging, surgical therapy, histopathological workup, and clinical follow-up were obtained. Histopathological examination included the pathological TNM staging, number of harvested and affected lymph nodes, extracapsular spread (ECS), grade of differentiation and depth of invasion (DOI). All cases were restaged according to the recent AJCC Cancer Staging Manual, 8th Edition [20]. Preoperative staging using CT or MRI scans and clinical examination determined the cN0 contralateral neck status. Clinical decision making for contralateral END was made for each patient according to the individual risk profile. Patients receiving checkpoint therapies were not included in this study. Clinical and radiographic follow-up examinations were performed according to the current guidelines [4] and retrospectively assessed to determine local and/or regional disease recurrence as well as the presence of distant metastasis. Information according to the individual death was assessed through patients charts and the institutional cancer registry of the Charité —Universitätsmedizin Berlin.

### 2.3. Statistical Analysis

Descriptive analysis was performed by using appropriate summary statistics, such as the mean and standard deviation for metric variables as well as median with interquartile ranges [25th; 75th percentile] for time periods, whereas categorical variables were presented as absolute and relative frequencies. Exploratory inferential statistics were conducted by using two-sided, two-sample t-tests in case of metric variables and two-sided chi-square tests in case of ordinal variables. Linear regression was used to determine the correlation of ipsilateral lymph node involvement and contralateral CLNM. Receiver-operating characteristic (ROC) curves were then used to determine cutoff values by Youden Index for the risk predictors of contralateral CLNM. Overall survival (OS), recurrence free survival (RFS), and neck control rate (NCR) were determined through Kaplan–Meier analysis. Survival curves were then statistically analyzed using the log rank test. *p*-values < 0.05 were considered statistically significant; however, due to the exploratory nature of the study, no adjustment for multiplicity was applied. Hence, all *p*-values are interpreted in a hypothesis-generating manner. All statistical analyses were conducted using SPSS version 28 (IBM Corp., Armonk, NY, USA).

## 3. Results

### 3.1. Patient Baseline Data

A total of 65 patients, 41 males (63.1%) and 24 females (36.9%), ranging in age from 34 to 86 years (mean 62.5 years; ±11.2), were included. Risk factors for OSCC were present in 38 cases (58.5%, regular use of tobacco and/or alcohol). As seen in Figure 2, the primary sites were as follows: the anterior two-thirds of the tongue (n = 24, 36.9%), the mandibular alveolar process (n = 13, 20.0%), the floor of the mouth (n = 12, 18.5%), the buccal mucosa (n = 7, 10.8%), the maxillary alveolar process (n = 2, 3.1%), the hard palate (n = 2, 3.1%), and multiple combined locations (n = 5, 7.7%). Preoperative staging of the ipsilateral neck was as follows: cN0 in 25 (38.5%) cases, cN1 in 16 (24.6%) cases, cN2a in 8 (12.3%) cases, and cN2b in 16 (24.6%) cases. Thus, the rate of occult CLNM in this cohort was 38.5%. Table 1 shows the distribution of baseline and demographic data between the groups. The number of patients with advanced staged tumors (cT3-4) was higher in group I as compared to group II. However, this difference was statistically not significant (^p^ = 0.15).

### 3.2. Lymph Node Management

Selective neck dissection (SND) of the cervical level I-III was performed for the ipsilateral site in 20 (30.8%) of all cases. In 45 patients (69.2%), an ipsilateral CND with the preservation of the spinal accessory nerve, the internal jugular vein and the sternocleidomastoid muscle was performed. For the contralateral cN0 neck, 16 patients (24.6%) received a SND (referred to as group I). In 49 (75.4%) patients, the unsuspicious contralateral neck was not addressed surgically (referred to as group II).

### 3.3. Histopathological Examination

The results from histopathological examination are shown in Table 2. There was no case of contralateral CLNM in group 1, resulting in 0% occult contralateral CLNM rate initially after surgery in contralateral cN0 cases. Data regarding tumors’ DOIs were available in 40 (61.5%) cases. The mean overall DOI was 11.1 mm (±6.7 mm).

### 3.4. Follow-Up and Disease Recurrence

Follow-up ranged from 6 to 114 months with a median time of 28 months [15.5;45.5]. Overall, 25 (38.5%) patients presented with local disease recurrence during follow-up, after a median time of 15 months [10.5;22]. In addition, 6 (9.2%) patients presented with secondary (recurrent) CLNM during follow-up, after a median time of 12.5 months [6.8;23.3]. In group I, 4 of 16 patients (25.0%) developed CLNM, while in group II, 2 of 49 (4.1%) developed CLNM.

### 3.5. Occurrence of Contralateral CLNM in Ipsilateral N+ Necks during Follow-Up

In 5 out of 6 CLNM cases (7.7% overall), the contralateral neck was affected, with 2 patients presenting with bilateral CLNM. According to the distribution of secondary CLNM overall, secondary contralateral CLNM was also more frequent in group I (18.8% versus 4.1%). In the 3 affected patients in group I, secondary CLNM was in 66.7% in cervical levels previously targeted by SND. In the remaining case, CLNM was present in level IV. In 4 of 5 (80.0%) patients with contralateral CLNM (group I: 2/16; group II 2/49), the region was covered by the target volume of adjuvant radiotherapy. The mean overall radiation dose in the neck was 50.5 Gy (±10.0). There was no statistically significant difference between both groups (*p* = 0.78). The overall median lymph node yield during neck dissection was 26 [19;36.5]. The ratio of ipsilateral positive lymph nodes to the overall removed nodes (lymph node ratio; LNR) during neck dissection correlated with the occurrence of contralateral CLNM (Coeff = 0.77 [−0.04;1.58], *p* = 0.07). The optimal cutoff point in predicting contralateral CLNM was 11.4% LNR (sensitivity: 50%; specificity: 83.1%).

### 3.6. Survival Analysis

The 3-year OS for both study groups was 65.8%. Secondary CLNM (3-year OS: 33.3% versus 68.5%, *p* = 0.15) and secondary contralateral CLNM reduced the OS overall (3-year OS: 30.0% versus 68.6%, *p* = 0.11). There were no significant differences in 3-year OS between the study groups (66.9% versus 66.1%, *p* = 0.59) (Figure 3). There was also no significant difference regarding 3-year RFS between both groups (54.4% versus 61.4%, *p* = 0.19). The three-year NCR was 91.3%. Group I presented with a significantly lower 3-year NCR than group II (79.8% versus 95.1%; *p* < 0.01).

## 4. Discussion

From a pathophysiological point of view, contralateral lymph node involvement can occur in different ways. Tumor spread across the midline with connection to efferent collateral lymphatic vessels is certainly one of the most common reasons for contralateral CLNM. In cases of lateralized OSCC without midline involvement, crossing afferent lymph vessels may also lead to contralateral spread [15]. Recent studies investigating the lymphatic drainage pattern using SLNB for lateralized OSCC revealed 12–20.6% bilateral/contralateral drainage [21,22]. However, the overall rate of occult contralateral metastasis was only approximately 2% in both studies. If there is no midline involvement, the ipsilateral pN+ stage and the number of affected lymph nodes present as independent risk predictors for contralateral CLNM [11,15]. The LNR, which represents the number of positive nodes relative to the total number of examined lymph nodes, serves as an independent prognostic item for OS and RFS in OSCC [23]. Different studies in the literature describe poor outcomes in follow-up of OSCC in patients with a high LNRE [24,25]. In the current study, the LNR correlated with the occurrence of contralateral CLNM. The determined threshold of 11.4% might be able to provide guidance in decision making after initial surgery and neck dissection about extension of neck clearance, extension of the field of adjuvant radiation, and closer follow-up than in the comparable low-risk profile group. Moreover, tumors’ depth of invasion is considered to be a risk predictor for CLNM overall, as has been reported in several studies [26,27]. For lateralized OSCC, Mahieu et al. reported that the DOI was also a predictor for contralateral CLNM, with mean DOIs of 8.5 mm (versus 5.9 mm) in patients with occult CLNM and 9.48 mm (versus 5.9 mm) in patients with contralateral regional recurrence [28]. Due to the advanced disease cohort in our study, our mean tumor DOI was 11.0 mm and did not correlate with contralateral CLNM. This also seems to be the case because the study group was heterogenous and tumors were in different locations within the oral cavity. DOI has been reported to differ between specific locations and should, therefore, not be regarded as a separate predictive item in this study group.

Elective neck dissection of the ipsilateral neck for OSCC is an established procedure and is still considered to be the gold standard due to the high risk of occult metastasis. The threshold of 20% is often referred to for the benefit of END versus observation only [29]. This benefit was also shown by D’Cruz et al. in a prospective trial emphasizing the role of END in contrast to a therapeutic approach in cases with CLNM during follow-up [10]. A remaining matter of debate is certainly the extend of the ND in cases with present ipsilateral nodal involvement. The nodal clearance of the cervical levels I to III according to Robbins, referred to as a SND, is considered to be the gold standard with CLNM located in levels Ia and Ib [4]. In cases with apparent CLNM in Levels IIa/b and III, the extension of the ND to levels IV and V as a prophylactic procedure is widely accepted as the standard of care. Clearance of all 5 levels under protection of the spinal accessory nerve, internal jugular vein, and the sternocleidomastoid muscle, is referred to as CND. However, it could be shown that CLNM is only rarely found in levels IV and V, even in pN+ necks [18]. This matter still raises the question of oncologic benefit of CND especially regarding peri- and postoperative morbidity of this extensive nodal clearance.

Nevertheless, the contralateral neck has not explicitly been addressed in a RCT and remains a subject of debate in cases of tumors not reaching the midline. Several studies investigated the risk of secondary contralateral CLNM in lateralized OSCC and reported CLNM rates between 2.9% and 7% for the untreated contralateral neck [29,30,31,32]. Unfortunately, these studies included not only pN+ patients but also pN0 in the same cohort. Three identified studies investigated the role of contralateral elective ND versus observation only [33,34,35]. Nobis et al. reported no significant difference between both groups in regard to OS and RFS [33]. However, the results from Nobis et al are difficult to compare with this study due to the fact that the authors only included early staged diseases (pT1-2). Consequently, AT was also only per-formed in 26.0% of the cases. 

Lim et al. reported a worse 5-year RFS in the END group versus observation only (68% versus 83%), which was similar to that in our study [34]. No contralateral neck recurrence was reported in this study, which included only bilateral cN0 patients. Knopf et al. also reported there was no statistically significant difference between groups [35]. It must be mentioned, however, that the bilateral END group had significantly more patients with adjuvant therapy and tumors reaching the midline. Singhavi et al. retrospectively investigated 78 patients who were treated by contralateral END after presenting with ipsilateral nodal recurrence during follow-up [36]. With a 23.1% rate of contralateral CLNM, END was reasonable in this cohort. In a meta-analysis comparing SND versus CND for patients with clinically suspicious necks, Liang et al presented five studies with contralateral neck recurrence rates between 0–13.9% [37]. It could be shown that in each of these five studies, the more extended CND even showed higher rates of contralateral neck recurrence.

The presented study group received bilateral radiation therapy of both neck sites. Patients with unilateral RT or those who refused or cancelled AT were excluded from the study. The results are therefore not only an argument concerning the surgical management of the contralateral neck site, but moreover the results underline the need for stringently performed AT. A current meta-analysis by Kaso et al. from 2021 investigated the rates of contralateral recurrence of OSCC patients, who received only ipsilateral adjuvant RT [38]. Analysis revealed a relatively low rate of contralateral neck recurrence of 3.4% (95% CI: 2.2‒5.4%). However, correlation to pN stages and localization of the primary tumor supported the recommendation to omit contralateral RT only in pN0-1 cases with well lateralized OSCC. Results of this study were supported by Al-Mamgani et al., who also advocate contralateral RT only according to T stages and midline involvement in oropharyngeal carcinomas [39]. However, patient selection and inclusion criteria of the current study explicitly only involved patients with bilateral RT according to local standards. De-escalation of AT and surgical neck management are two parameters that should be adjusted and analyzed apart in randomized clinical trials.

In the current study, patients with contralateral ND had a significantly lower NCR than patients without contralateral ND. This finding suggests that extended neck treatment ensures less neck control. To our knowledge, this finding was not presented elsewhere in the literature so far. One explanation could be the higher distribution of patients with advanced stage diseases and local recurrence in group I. However, these differences are statistically not significant. Due to the low sample size, especially in group I, this finding may be biased, and must be considered with caution.

The current study was able to show the low incidence of contralateral CLNM in ipsilateral node positive OSCC. Nevertheless, this investigation had some limitations, including the relatively low number of included patients in a retrospective setting with contralateral CLNM during follow-up. Therefore, a risk prediction model by multivariate regression was statistically inconclusive, and statistical significance could not be established. A larger sample size and a more extended follow-up period might have had an influence on the results. 

The main advantages of the study were the defined inclusion criteria and the ho-mogenous composition of the study groups according to similar pathological characteristics and similar adjuvant treatment.

## 5. Conclusions

All in all, the results of this study suggest that in cases of lateralized OSCC with ipsilateral nodal disease, the contralateral neck should not be treated by END, also in order to protect regional immune competence. This conclusion is supported by the facts that (i) there were initially no cases of occult contralateral CLNM in this cohort, (ii) the rate of secondary contralateral CLNM during follow-up was only 7.7% and (iii) there was no survival benefit in patients treated by contralateral END versus adjuvant radiotherapy only. However, the extent of ipsilateral nodal involvement increases the risk for contralateral CLNM, with a threshold of 11.4% LNR identified in this study. This fact may guide the surgeon toward an extension of ND to the contralateral neck or toward extended adjuvant treatment during case discussions in interdisciplinary tumor conferences.

## Figures and Tables

**Figure 1 cancers-15-01088-f001:**
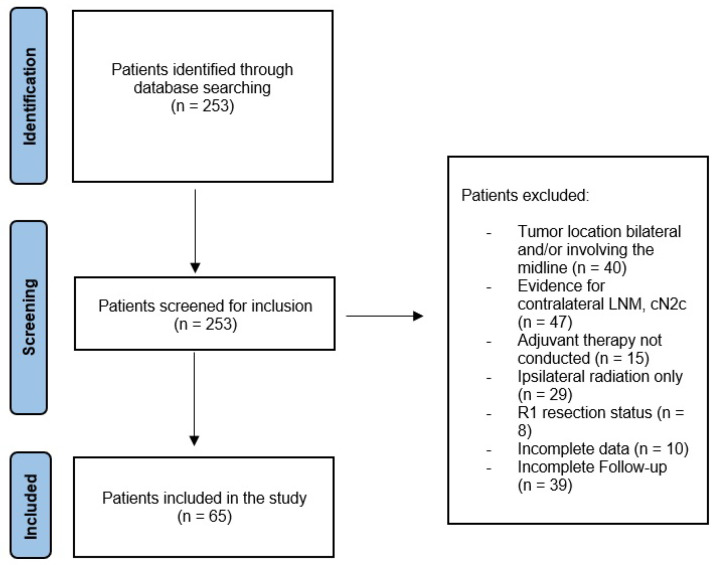
Flowchart demonstrating the process of inclusion.

**Figure 2 cancers-15-01088-f002:**
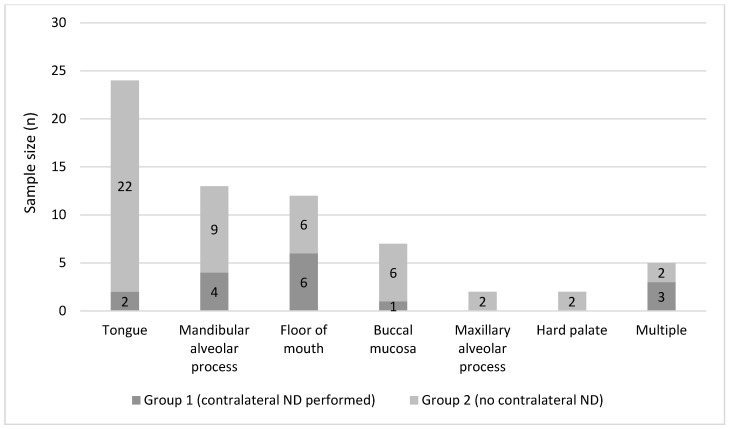
Distribution of primary tumor sites between both groups.

**Figure 3 cancers-15-01088-f003:**
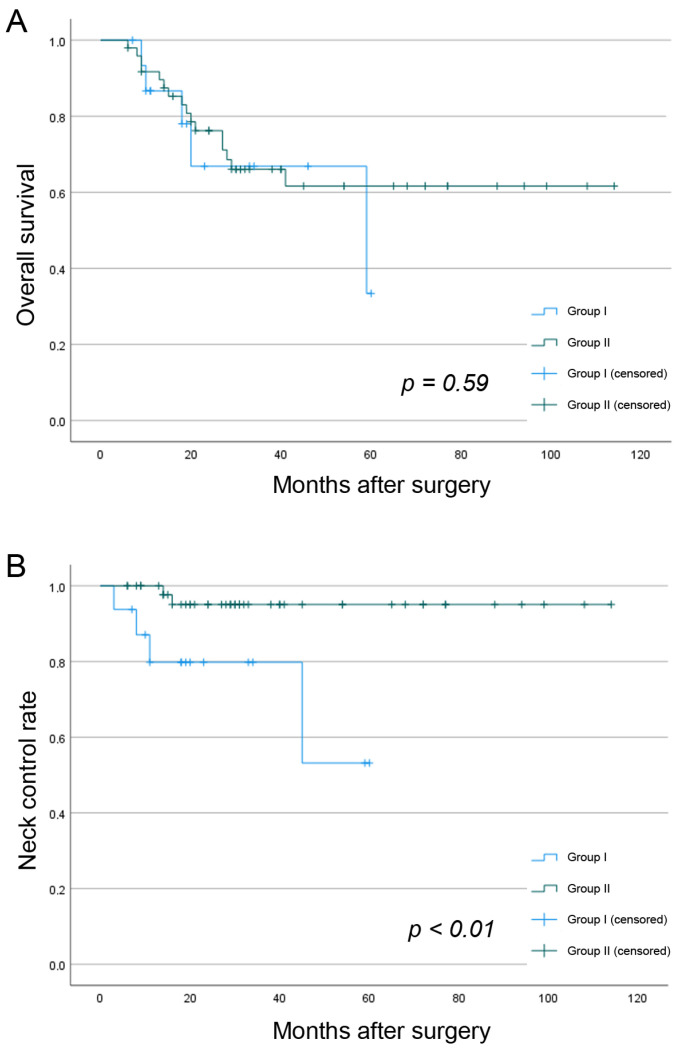
Kaplan–Meier survival curves demonstrating the OS (**A**) and NCR (**B**) between the study groups.

**Table 1 cancers-15-01088-t001:** Baseline and demographic patient data.

	Group I (Contralateral ND; n = 16, 24.6%)	Group II (No Contralateral ND; n = 49, 75.4%)	*p*-Value
Gender			0.59
Female	5 (31.3%)	19 (38.8%)	
Male	11 (68.7%)	30 (61.2%)	
Age in years (mean)	62.9 (±7.0)	62.4 (±12.3)	0.89
Alcohol & Tobacco			0.12
Yes	12 (75.0%)	26 (53.1%)	
No	4 (25.0%)	23 (46.9%)	
cT Stage			0.15
cT1–2	7 (43.8%)	32 (65.3%)	
cT3–4	9 (56.2%)	17 (34.7%)	
Ipsilateral cN Stage			0.93
cN0	7 (43.8%)	18 (36.7%)	
cN1	4 (25.0%)	12 (24.5%)	
cN2a	2 (12.5%)	6 (12.2%)	
cN2b	3 (18.8%)	13 (26.5%)	

**Table 2 cancers-15-01088-t002:** Histopathological examination.

	Group I (Contralateral ND; n = 16, 24.6%)	Group II (No Contralateral ND; n = 49, 75.4%)	*p*-Value
pT Stage			0.52
pT1–2	7 (43.8%)	26 (53.1%)	
pT3–4	9 (56.2%)	23 (46.9%)	
pN Stage			
pN1	9 (56.2%)	12 (24.5%)	
pN2a	0	0	
pN2b	2 (12.5%)	22 (44.9%)	
pN2c	0	0	
pN3a	0	0	
pN3b	5 (31.3%)	15 (30.6%)	
ECS			0.96
Yes	5 (31.3%)	15 (30.6%)	
No	11 (68.7%)	34 (69.4%)	
Grade of differentiation			0.46
G1	0	2 (4.1%)	
G2	14 (87.5%)	36 (73.5%)	
G3	2 (12.5%)	11 (22.4%)	
Depth of invasion (mean)	12.0 (±7.4)	10.8 (±6.6)	0.57

Depth of invasion is indicated as mm.

## Data Availability

All data is available in this article.

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
