# Peer review of "Management of the Contralateral Neck in Unilateral Node-Positive Oral Squamous Cell Carcinoma"

_cancers, 2023, doi:10.3390/cancers15041088_

Round 1

Reviewer 1 Report

3. Results

3.2.  Lymphe node management

There is a confusion in the nomenclature of neck dissection (SND and MRND) please use the current classification (SND and CND - see in NCCN guideline)

Explanation is needed how the surgical candidates were selected for surgery, what specific criterias were used ?

Author Response

1) Thank you for this comment. We adapted the nomenclature of neck dissection according to the current NCCN guidelines.

2) Thank you for pointing this out. The decision for contralateral elective neck dissection was made individually for patients with an assumed higher risk profile for contralateral CLNM. This fact was mainly supported by the cT-staging, which is now also included in the article. In our study, patients receiving contralateral ND had a higher rate of advanced-stage diseases (cT3-4). However, this difference is statistically not significant (56.2% in group I versus 34.7% in group II, p = 0.15).

Reviewer 2 Report

Especially from a clinical point of view, the presented study is of high interest to a wide readership in the field. Although the study population is quite small, the authors were able to address the question of the study in an appropriate way. Besides some minor issues with wording and language, the following minor points should be addressed by the authors:

1) The percentage of noxious substance abuse is quite low in the study population. Were there also p16-positive cases included? Of course, here would have been a higher incidence rate of nodal involvement.

2) The type of adjuvant therapies was not mentioned, were there also checkpoint therapies included?

3) The authors should also discuss the findings of Fig. 3B

All together a sound story with is of high relevance. After these minor changes, I support publication of the present manuscript!

Author Response

Dear reviewer, 

Thank you for the comments on our manuscript. In this following section, we provide a point-by-point response to each question posed or comment made:

1) In the past years, the p16 status was not routinely determined in our department for OSCC. Therefore, we did not included it to this article.

2) Thank you for this comment. As mentioned in our inclusion criteria, we analysed patients with newly diagnosed OSCC who were surgically treated in a curative setting. All of these patients received adjuvant radiotherapy with or without chemotherapy. Patients receiving checkpoint therapies were not included in this study.

3) Thank you for pointing this out. In our study, patients without contralateral ND had a higher neck control rate than patients with contralateral ND. The reason for this cannot be concluded from our data. Due to the low sample size in group I, this fact must be considered with caution since it may be biased. One explanation may be the higher distribution of patients with advanced stage diseases (56.2% vs. 46.9%, p = 0.52) as well as local recurrence (43.8 vs. 38.3%; p = 0.43) in group I. However, these differences are statistically not significant.

The following sections were incorporated into the manuscript:

“Overall, 25 (38.5%) patients presented with local disease recurrence during follow-up (group I: 43.8% versus group II: 38.3%, p = 0.43), after a median time of 15 months [10.5;22].”

“In the current study, patients with contralateral ND had a significantly lower NCR than patients without contralateral ND. This finding suggests that extended neck treatment ensures less neck control. To our knowledge, this finding was not presented elsewhere in the literature so far. One explanation could be the higher distribution of patients with advanced stage diseases and local recurrence in group I. However, these differences are statistically not significant. Due to the low sample size, especially in group I, this finding may be biased, and must be considered with caution.”

Reviewer 3 Report

The authors performed a retrospective single institution analysis on contralateral elective neck dissection in oral squamous cell carcinoma patients with positive ipsilateral neck disease. Overall, while a limited overall cohort number, I find this a well-written study, addressing a useful clinical question. I do have some comments/suggestions:

1. Did the authors include clinical T stage in their demographics and analysis? This would be important to account for initial clinical decision-making and additional factors for occult contralateral neck metastasis.

2. Patients with a follow-up time less than 2 years (e.g. authors note follow-up of 6 months) may not have adequate time for assessment of disease recurrence. Is there a reason the authors made this minimum interval follow-up they could help justify, rather than a longer established follow-up? This would have implications in their survival analyses as well.

Author Response

Dear reviewer, 

Thank you for the valuable comments on our manuscript. Here we provide a point-by-point response to each question posed or comment made

1) Thank you for pointing this out! After reviewing the cT stage we could show that more patients with advanced stage diseases (cT3-4) received contralateral ND (56.2% versus 34.7%, p = 0.15). This may be one of the reasons which guided clinical decision making towards neck management.  

2) Thank you for this comment. Further restriction of the minimal follow-up time would result in such a lower sample size, especially in group I, that statistical analysis would not make sense anymore. Therefore, we made the decision for 6 months in order to balance out sample size and follow-up time. We agree that a minimum of 18-24 months would be ideal. However, this would exclude 19 patients in this study (7/16 in group I).